# Intraoperative Positive Pancreatic Parenchymal Resection Margin: Is It a True Indication of Completion Total Pancreatectomy after Partial Pancreatectomy for Pancreatic Ductal Adenocarcinoma?

**Ji-Hye Jung, So-Jeong Yoon, Ok-Joo Lee**  **, Sang-Hyun Shin, Jin-Seok Heo and In-Woong Han** *

Division of Hepatobiliary-Pancreatic Surgery, Department of Surgery, Samsung Medical Center, Sungkyunkwan University School of Medicine, Seoul 06351, Korea; soghei@amc.seoul.kr (J.-H.J.); sojeong.yoon@samsung.com (S.-J.Y.); 106254@schmc.ac.kr (O.-J.L.); surgeonssh@skku.edu (S.-H.S.); jinseok.heo@samsung.com (J.-S.H.)

* Correspondence: iw.han@samsung.com; Tel.: +82-2-3410-0772; Fax: +82-2-3410-6980

**Abstract:** Background: Total pancreatectomy (TP) can be performed in cases with positive resection margin after partial pancreatectomy for pancreatic cancer. However, despite complete removal of the residual pancreatic parenchyme, it is questionable whether an actual R0 resection and favorable survival can be achieved. This study aimed to identify the R0 resection rate and postoperative outcomes, including survival, following completion TP (cTP) performed due to intraoperative positive margin. Methods: From 1995 to 2015, 1096 patients with pancreatic ductal adenocarcinoma underwent elective pancreatectomy at the Samsung Medical Center. Among these, 25 patients underwent cTP, which was converted during partial pancreatectomy because of a positive resection margin. To compare survival after R0 resection between the cTP R0 and pancreaticoduodenectomy (PD) R0 cases, propensity score matching was conducted to balance the baseline characteristics. Results: The R0 rate of cTP performed due to intraoperative positive margin was 84% (21/25). The overall 5-year survival rate (5YSR) in the 25 cTP cases was 8%. There was no difference in the 5YSR between the cTP R0 and cTP R1 groups (9.5% versus 0.0%, $p = 0.963$). However, the 5YSR of the cTP R0 group was significantly lower than that of the PD R0 group (9.5% versus 20.0%, $p = 0.022$). There was no distinct difference in postoperative complications between the cTP R0 versus cTP R1 and cTP R0 versus PD R0 groups. Conclusions: In cases with intraoperative positive pancreatic parenchymal resection margin, survival after cTP was not favorable. Careful patient selection is needed to perform cTP in such cases.

**Keywords:** completion total pancreatectomy (cTP); R0 resection; R1 resection; pancreaticoduodenectomy (PD); postoperative outcomes

## 1. Introduction

To achieve improved outcomes in pancreatic cancer treatment, resection of the primary tumor and lymphadenectomy is the mainstay of curative treatment [1–3]. The survival rate is higher in patients who underwent surgery than in those who did not undergo surgery, which has been reported in many studies [4–6]. In addition to the implementation of surgery, it is important to secure radicality through confirming finally tumor-negative resection margin for improved survival outcomes, which has also been reported in many studies [7–10]. It can be said that achieving a tumor-negative margin is still the primary purpose of pancreatic cancer surgery. This is the main reason for conversion of partial pancreatectomy, such as pancreaticoduodenectomy (PD) or distal pancreatectomy (DP), to total pancreatectomy (TP) due to a positive pancreatic parenchymal resection margin during the operation. Despite the major disadvantages of TP, including life-long endocrine and exocrine insufficiency, TP has been steadily performed since favorable long-term outcomes

were obtained by radical resection and short-term outcomes were deemed tolerable [11–15]. Moreover, postoperative outcomes after TP have been reported to be relatively equivalent to those after PD [16–19].

However, although the whole pancreatic parenchyme is completely resected due to intraoperative positive pancreatic margin, whether an actual R0 resection and higher survival are achieved is questionable. When making a pathological evaluation of the resected specimen, not only transection margin but also circumferential margin are examined, depending on the tumor location and the type of surgery performed [9,20]. There may be cases in which the pancreatic parenchymal margin, mainly the pancreatic duct margin, is negative; however, other margins such as retropancreatic margin, superior mesenteric vein (SMV) margin, and superior mesenteric artery (SMA) margin are involved by tumor and concluded to be positive.

Studies on the relationship between the resection margin status and survival in patients who underwent partial pancreatectomy, such as PD or DP, have been conducted, and they reported no significant associations [21–25]. Moreover, several studies have reported that the benefit of additional pancreatic resection in the situation of a positive frozen biopsy during partial pancreatectomy did not have much impact on survival [26,27]. It appears that there have been few studies on margin status and survival limited to TP [23]. We aimed to study the postoperative outcomes and the value of completion TP (cTP) performed to obtain R0 resection in the case of intraoperative positive margin; hence, we investigated cases in which conversion to cTP was performed as an intraoperative decision due to a positive pancreatic parenchymal margin.

## 2. Materials and Methods

### 2.1. Patients and Data Collection

After approval from the Institutional Review Board (IRB) (No. 2019-07-150), we searched consecutive patients who underwent elective pancreatectomy for pancreatic ductal adenocarcinoma at Samsung Medical Center between 1995 and 2015. Figure 1 shows the inclusion and exclusion criteria of our study. There were 1096 cases of elective pancreatectomy in total; 632 and 378 cases of PD and DP, respectively, and 86 cases of TP. In our institute, we usually transect the pancreas at the left border of SMV. The location of pancreas transection is adjusted according to the tumor extent. The pancreatic duct transection margin was evaluated routinely by intraoperative frozen biopsy. A total of 25 cases that were converted to cTP due to positive pancreatic parenchymal resection margin confirmed by frozen section biopsy were collected. In the 25 cTP cases, the initially planned procedures were PD for 20 cases and DP for 5 cases. We identified whether R0 resection was achieved after resecting the remnant whole pancreas to compare these cases with cases where R0 resection was not achieved despite performing cTP. In the final permanent pathological report, not only the pancreatic transection margin but also the SMA margin is routinely determined. It was reported as the retropancreatic margin prior to around 2012. When any one of these margins is confirmed to have been invaded by tumor, it is determined as R1. Since about 2012, we redefined the R1 to be within 1 mm of the distance between the tumor and the resection margin [28,29]. In almost all cases (21/25), radicality was achieved after resecting the entire pancreas. However, four out of 25 cases were nonetheless confirmed as positive in the final pathological evaluation results; therefore, they were concluded as R1 resection. The clinical, operative, and pathological characteristics and postoperative outcomes, including overall survival, were compared between 21 cases of R0 resection and four cases of R1 resection.

In cases where R0 resection was achieved through cTP, we investigated whether there were any differences compared to the PD R0 case, especially in terms of overall survival. Among the 632 patients with PD, 445 were identified as R0 resection in the final pathology. The discrepancy in the number of patients between the two groups was too large to compare 445 patients in the PD R0 case with 21 patients in the cTP R0 case. Therefore, we conducted propensity score matching (PSM) analysis to extract balanced

cases. Eight factors that can affect survival were used for the matching factor: age, sex, American Society of Anesthesiologists (ASA) score, T stage, N stage, perineural invasion (PNI), lymphovascular invasion (LVI), and whether postoperative adjuvant treatment was performed. As a result of PSM, 40 cases in the PD R0 group were generated, and various perioperative characteristics, including overall survival, were compared between the 21 cases in the cTP R0 group and 40 cases in the PD R0 group.

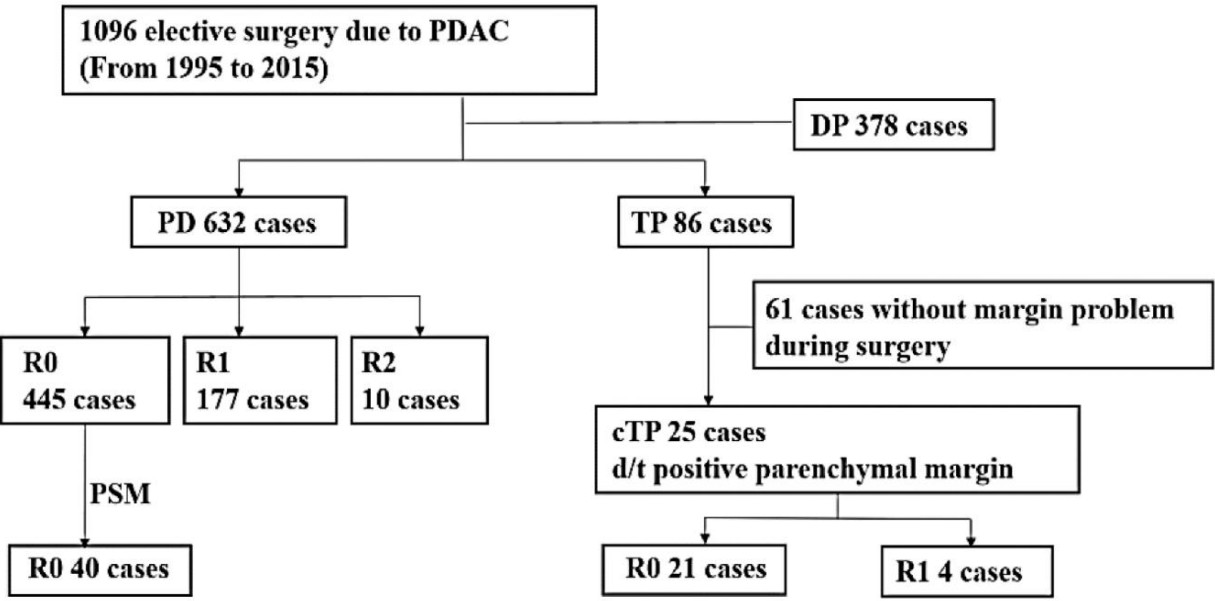

**Figure 1.** Inclusion and exclusion flows in our study. Abbreviations: PDAC, pancreatic ductal adenocarcinoma; PD, pancreaticoduodenectomy; TP, total pancreatectomy; DP, distal pancreatectomy; cTP, completion total pancreatectomy; d/t, due to; PSM, propensity score matching.

### 2.2. Statistical Analysis

Statistical analyses to compare clinical, operative, and pathological characteristics and postoperative outcomes were conducted using the IBM SPSS statistical software, version 27 (Chicago, IL, USA). Continuous variables between the two groups were compared using the independent t-test, and categorical data were analyzed using the chi-square test. The overall survival rate was estimated, and the curves were plotted using the Kaplan–Meier method. As mentioned previously, we conducted PSM to balance the PD R0 group and the cTP R0 group using eight factors. To execute the PSM, we used the SAS statistical software (version 9.4; SAS Institute, Cary, NC, USA) and the R statistical software (version 4.0.0; Vienna, Austria; http://www.R-project.org/ accessed on 24 April 2020). We used the nearest neighbor matching method with a caliper of 0.25 for all patients. Eight variables, including age, sex, ASA score, T stage, N stage, PNI, LVI, and whether postoperative adjuvant treatment was performed were applied for PSM factors. As a result, the most balanced cases were extracted. Differences with probability (*p*) value 0.05 or less were considered statistically significant in all analyses.

### 3. Results

#### 3.1. Clinical, Operative, and Pathological Characteristics

The R0 rate of cTP performed due to intraoperative positive pancreatic parenchymal resection margin was 84% (21/25). Of the four cases that were determined to be R1 resection in the final permanent pathology despite performing cTP, the retropancreatic margin was invaded by the tumor in two cases. In the other two cases, the SMA margin was confirmed to be invaded.

Table 1 shows the results of comparing the clinical and operative characteristics of the cTP R0 and cTP R1 groups. There were no significant differences in most factors, including

age, sex, BMI, ASA scores, preoperative CA19-9 levels, and neoadjuvant treatment. In the cTP R1 group compared with the cTP R0 group, the rate of combined vascular surgery was higher, the operation time was longer, and the estimated blood loss (EBL) was greater; however, all were not significant. All the vascular surgeries performed together were cases in which the PV or SMV was invaded by the tumor; therefore, resection and anastomosis (R&A) were performed. No other organs were resected to achieve radicality in either group. When comparing the pathological characteristics such as T stage, N stage, and degree of differentiation, there was no significant difference between the two groups. Contrary to expectations, the cTP R1 group did not have significantly advanced stage.

**Table 1.** Comparison of clinical and operative characteristics between R0 cases and R1 cases after cTP (number (percent) and mean ± standard deviation).

|  | cTP R0 (*n* = 21) | cTP R1 (*n* = 4) | *p*-Value |
|---|---|---|---|
| Age (median, years) | 62.0 | 71.5 | 0.144 |
| Sex |  |  | 0.656 |
| Male | 13 (61.9) | 2 (50.0) |  |
| Female | 8 (38.1) | 2 (50.0) |  |
| BMI (kg/m$^2$) | 23.0 ± 2.8 | 21.9 ± 1.4 | 0.499 |
| Preoperative DM |  |  | 0.238 |
| No | 9 (42.9) | 3 (75.0) |  |
| Yes | 12 (57.1) | 1 (25.0) |  |
| ASA score |  |  | 0.367 |
| 1 | 1 (4.8) | 1 (25.0) |  |
| 2 | 19 (90.4) | 3 (75.0) |  |
| 3 | 1 (4.8) | 0 (0.0) |  |
| PreOP albumin (g/dL) | 4.1 ± 0.4 | 4.1 ± 0.3 | 0.779 |
| PreOP total protein (g/dL) | 6.9 ± 0.5 | 6.7 ± 0.7 | 0.470 |
| PreOP CA19-9 (U/mL) |  |  | 0.524 |
| 37 or below | 7 (33.3) | 2 (50.0) |  |
| Higher than 37 | 14 (66.7) | 2 (50.0) |  |
| PreOP total bilirubin (mg/dL) | 2.8 ± 3.7 | 6.2 ± 7.2 | 0.163 |
| PreOP biliary drainage |  |  | 0.656 |
| No | 13 (61.9) | 2 (50.0) |  |
| Yes | 8 (38.1) | 2 (50.0) |  |
| Neoadjuvant treatment |  |  | - |
| No | 21 (100.0) | 4 (100.0) |  |
| Yes | 0 (0.0) | 0 (0.0) |  |
| Other organ resection |  |  | - |
| No | 21 (100.0) | 4 (100.0) |  |
| Yes | 0 (0.0) | 0 (0.0) |  |
| Combined vascular surgery |  |  | 0.184 |
| No | 17 (81.0) | 2 (50.0) |  |
| Yes | 4 (19.0) | 2 (50.0) |  |
| Operation time (minutes) | 366.9 ± 57.8 | 456.0 ± 185.1 | 0.408 |
| Estimated blood loss (mL) | 728.6 ± 398.0 | 825.0 ± 623.8 | 0.688 |

Abbreviations: cTP, completion total pancreatectomy; BMI, body mass index; DM, diabetes mellitus; ASA, American society of anesthesiologists; PreOP, preoperative; CA 19-9, carbohydrate antigen 19-9.

The results of comparing the clinical and operative characteristics between the cTP R0 group and the PD R0 group extracted by PSM are shown in Table 2. In the PD R0 group, as compared to the cTP R0 group, the levels of preoperative albumin and total protein were lower and the preoperative total bilirubin level was higher, resulting in a higher rate of preoperative biliary drainage insertion. Except for these, there were no significant differences in any of the factors, such as age, sex, CA19-9 level, neoadjuvant treatment, and combined vascular surgery. In particular, the operation time and EBL were comparable between the groups. There was no significant difference in the pathological characteristics

between the two groups, which are presented in Table 3. Although the cTP R0 group had, on average, larger tumors than the PD R0 group, the difference was not significant.

**Table 2.** Comparison of clinical and operative characteristics between cTP R0 group and PD R0 group extracted by PSM (number (percent) and mean ± standard deviation).

|  | cTP R0 ($n = 21$) | PD R0 ($n = 40$) | $p$-Value |
|---|---|---|---|
| Age (median, years) | 62.0 | 60.5 | 0.893 |
| Sex |  |  | 0.885 |
| Male | 13 (61.9) | 24 (60.0) |  |
| Female | 8 (38.1) | 16 (40.0) |  |
| BMI (kg/m$^2$) | 22.9 ± 2.8 | 22.7 ± 3.9 | 0.471 |
| Preoperative DM |  |  | 0.202 |
| No | 9 (42.9) | 24 (60.0) |  |
| Yes | 12 (57.1) | 16 (40.0) |  |
| ASA score |  |  | 1.000 |
| 1 | 1 (4.8) | 2 (5.0) |  |
| 2 | 19 (90.4) | 36 (90.0) |  |
| 3 | 1 (4.8) | 2 (5.0) |  |
| PreOP albumin (g/dL) | 4.1 ± 0.4 | 3.9 ± 0.4 | 0.025 |
| PreOP total protein (g/dL) | 6.9 ± 0.5 | 6.5 ± 0.5 | 0.006 |
| PreOP CA19-9 (U/mL) |  |  | 0.789 |
| 37 or below | 7 (33.3) | 12 (30.0) |  |
| Higher than 37 | 14 (66.7) | 28 (70.0) |  |
| PreOP total bilirubin (mg/dL) | 2.8 ± 3.7 | 5.9 ± 6.2 | 0.011 |
| PreOP biliary drainage |  |  | 0.016 |
| No | 13 (61.9) | 12 (30.0) |  |
| Yes | 8 (38.1) | 28 (70.0) |  |
| Neoadjuvant treatment |  |  | 0.465 |
| No | 21 (100.0) | 39 (97.5) |  |
| Yes | 0 (0.0) | 1 (2.5) |  |
| Other organ resection |  |  | 0.198 |
| No | 21 (100.0) | 37 (92.5) |  |
| Yes | 0 (0.0) | 3 (7.5) |  |
| Combined vascular surgery |  |  | 1.000 |
| No | 17 (81.0) | 33 (82.5) |  |
| Yes | 4 (19.0) | 7 (17.5) |  |
| Operation time (minutes) | 366.9 ± 57.8 | 340.7 ± 61.1 | 0.075 |
| Estimated blood loss (mL) | 728.6 ± 398.0 | 732.8 ± 629.2 | 0.225 |

Abbreviations: cTP, completion total pancreatectomy; PD, pancreaticoduodenectomy; BMI, body mass index; DM, diabetes mellitus; ASA, American society of anesthesiologists; PreOP, preoperative; CA 19-9, carbohydrate antigen 19-9.

**Table 3.** Comparison of pathological characteristics between cTP R0 group and PD R0 group extracted by PSM (number (percent) and mean ± standard deviation).

|  | cTP R0 (*n* = 21) | PD R0 (*n* = 40) | *p*-Value |
|---|---|---|---|
| T stage |  |  | 0.964 |
| T2 | 13 (61.9) | 25 (62.5) |  |
| T3 | 8 (38.1) | 15 (37.5) |  |
| Tumor size (cm) | 4.5 ± 3.4 | 3.8 ± 1.3 | 0.206 |
| N stage |  |  | 0.919 |
| N0 | 7 (33.3) | 14 (35.0) |  |
| N1 | 10 (47.6) | 17 (42.5) |  |
| N2 | 4 (19.1) | 9 (22.5) |  |
| Harvested LN | 28.6 ± 14.5 | 22.7 ± 10.9 | 0.080 |
| Metastatic LN | 2.0 ± 2.5 | 3.1 ± 5.8 | 0.732 |
| M stage |  |  | - |
| M0 | 21 (100.0) | 40 (100.0) |  |
| M1 | 0 (0.0) | 0 (0.0) |  |
| Differentiation |  |  | 0.591 |
| Well | 1 (4.8) | 5 (12.5) |  |
| Moderate | 12 (57.1) | 21 (52.5) |  |
| Poor | 7 (33.3) | 9 (22.5) |  |
| Unknown | 1 (4.8) | 5 (12.5) |  |
| PNI |  |  | 1.000 |
| No | 0 (0.0) | 1 (2.5) |  |
| Yes | 15 (71.4) | 27 (67.5) |  |
| Unknown | 6 (28.6) | 12 (30.0) |  |
| LVI |  |  | 1.000 |
| No | 1 (4.8) | 3 (7.5) |  |
| Yes | 7 (33.3) | 11 (27.5) |  |
| Unknown | 13 (61.9) | 26 (65.0) |  |

Abbreviations: cTP, completion total pancreatectomy; PD, pancreaticoduodenectomy; PSM, propensity score matching; LN, lymph node; PNI, perineural invasion; LVI, lymphovascular invasion.

### 3.2. Postoperative Outcomes

Between the cTP R0 group and the cTP R1 group, most of the postoperative short-term outcomes that occurred within 90 days after surgery did not show any difference. Length of stay, postoperative ICU (intensive care unit) stay, in-hospital stay, 90-day mortality, adjuvant treatment, and others were not significantly different between the two groups. The overall 5-year survival rate (5YSR) of the 25 cTP cases was 8.0%. Figure 2 shows that there was no significant difference in the overall survival regardless of whether radical clearance was achieved or determined to be R1 resection, even after cTP was performed due to positive pancreatic parenchymal margin (5YSR 9.5% versus 0.0% and median survival 13.0 months versus 9 months; *p* = 0.963).

A comparison of the postoperative short-term outcomes between the cTP R0 group and the PD R0 group extracted by PSM is presented in Table 4, showing no significant difference in all complications. Adjuvant treatment was not significantly different between the cTP R0 group and the PD R0 group. In contrast to short-term outcomes, there was a significant difference in the overall survival between the cTP R0 group and the PD R0 group, unlike the comparison between the cTP R0 group and the cTP R1 group. Figure 3 shows that the overall survival was significantly higher in the PD R0 group (5YSR 20.0% versus 9.5% and median survival 21.0 months versus 13.0 months; *p* = 0.022).

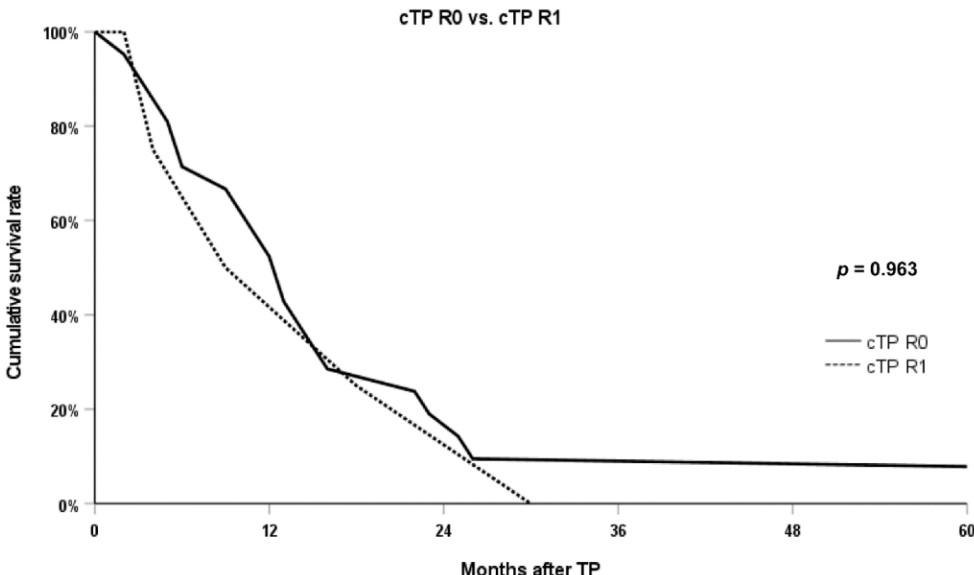

**Figure 2.** Comparison of overall survival between R0 cases and R1 cases after cTP. Abbreviations: cTP, completion total pancreatectomy.

**Table 4.** Comparison of postoperative outcomes between cTP R0 group and PD R0 group extracted by PSM (number (percent) and mean $\pm$ standard deviation).

|  | cTP R0 ($n$ = 21) | PD R0 ($n$ = 40) | $p$-Value |
|---|---|---|---|
| Clavien-Dindo classification |  |  | 0.323 |
| No complication | 15 (71.4) | 26 (65.0) |  |
| I | 1 (4.8) | 3 (7.5) |  |
| II | 1 (4.8) | 7 (17.5) |  |
| IIIa | 3 (14.2) | 1 (2.5) |  |
| IIIb | 1 (4.8) | 1 (2.5) |  |
| IVa | 0 (0.0) | 2 (5.0) |  |
| IVb | 0 (0.0) | 0 (0.0) |  |
| V | 0 (0.0) | 0 (0.0) |  |
| Length of stay (days) | 16.1 $\pm$ 6.3 | 16.5 $\pm$ 15.9 | 0.051 |
| Postoperative ICU stay (days) | 1.9 $\pm$ 2.0 | 1.2 $\pm$ 1.0 | 0.151 |
| In-hospital mortality | 0 (0.0) | 0 (0.0) | - |
| 90-day mortality | 1 (4.8) | 0 (0.0) | 0.344 |
| Re-admission within 90 days | 0 (0.0) | 2 (5.0) | 0.541 |
| Adjuvant treatment |  |  | 0.993 |
| No | 10 (47.6) | 19 (47.5) |  |
| Yes | 11 (52.4) | 21 (52.5) |  |

Abbreviations: cTP, completion pancreatectomy; PD, pancreaticoduodenectomy; PSM, propensity score matching; ICU, intensive care unit.

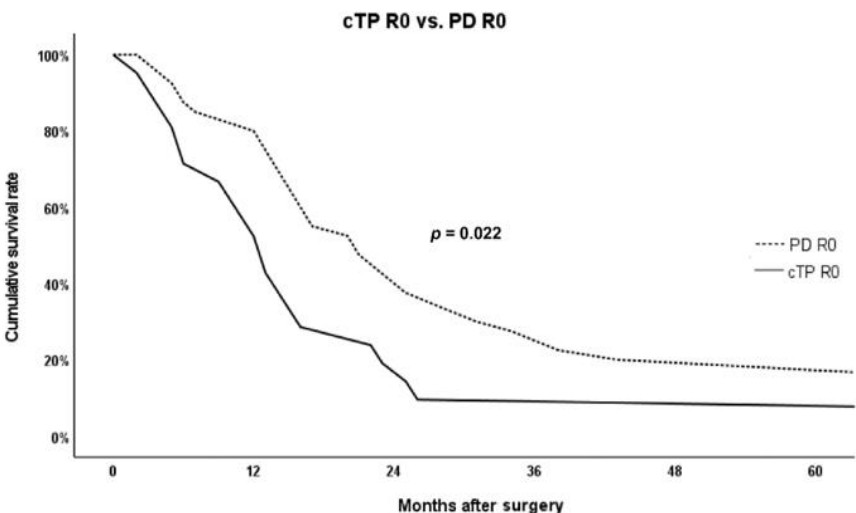

**Figure 3.** Comparison of overall survival between cTP R0 group and PD R0 group extracted by PSM. Abbreviations: cTP, completion total pancreatectomy; PD, pancreaticoduodenectomy; PSM, propensity score matching.

## 4. Discussion

In cases where the surgical decision was converted and cTP was performed because tumor invasion of the pancreatic parenchymal resection margin was confirmed through frozen section biopsy during partial pancreatectomy, it would be worthwhile to study the perioperative outcomes, including short-term postoperative complications and long-term prognosis. In our center, a total of 86 TP cases were performed for pancreatic ductal adenocarcinoma in about 20 years, and among them, there were 25 cases that were confirmed as positive pancreatic resection margin from frozen biopsy during the surgery and the surgical decision was converted to performing cTP. These numbers appear to be relatively low, even in our high-volume center.

There was no significant difference in the overall survival between the 21 cases and four cases finally identified as R0 resection and R1 resection, respectively. This indicated that even though radical clearance was achieved after cTP in cases of intraoperative positive pancreatic parenchymal resection margin, the long-term prognosis did not improve significantly. R0 resection is widely accepted as a common endpoint of surgical oncology in most surgically managed malignancies. However, the results of our study do not support this principle. It is thought to suggest that a positive margin status at the time of operation would be an indicator of that the tumor already had highly aggressive biology, resulting in early recurrence or systemic metastasis [26,30].

We also extracted the cases of PD that were balanced with the 21 cases in which R0 resection was achieved through cTP and compared various characteristics between the two groups. There was no significant difference in the postoperative short-term outcomes between the two groups. The cTP did not result in an unusually longer hospital/ICU stay or fatal complications. It can be said that cTP, performed when oncologically indicated, can be equivalent to PD in terms of safety. On the contrary, in terms of long-term outcomes, it was identified that although the whole pancreas was removed to secure radicality, the survival rate did not reach that of the PD group. In the situation of intraoperative positive resection margin, it was likely that micro-metastasis already existed even though radical clearance was achieved by performing cTP. Pancreatic cancer has been accepted as a systemic disease with aggressive biology [1,31,32]. When selecting the PD R0 cases that were balanced with the cTP R0 cases through PSM analysis, not only T stage, N stage, and postoperative adjuvant treatment but also PNI and LVI were used as matching factors to control those factors, as these are representative factors known to influence survival [33–35]. Nevertheless, the result that survival of the cTP R0 group did not reach that of the PD

R0 group increases the possibility of the presence of micro-metastasis in a condition that required cTP.

When faced with a situation of intraoperative positive margin after partial pancreatectomy, additional resection is performed and positive margin is still reported in frozen biopsy, so conversion to cTP can be considered. And cTP can be considered immediately without additional resection. In any case, the intraoperative decision of whether to perform cTP should be made carefully while taking into account that R0 resection may not be achieved despite cTP due to circumferential margin and that cTP has an inferior survival rate compared to PD.

Several complications such as diabetes mellitus, exocrine insufficiency, and liver disease occur after TP. These long-term complications must be managed throughout life and have the potential to affect survival [36,37]. However, these data were not investigated in our study, and therefore, it cannot be excluded that they may have contributed to the low survival of the cTP R0 group. This is a limitation of our study.

Another thing to discuss is that the PD R0 group was compared to the cTP R0 group. It would have been a more direct comparison if we had compared the cTP R0 group with the cases that had a positive frozen biopsy result during PD but did not achieve negative margin through additional resection. However, we could not find those cases in our center. In other words, when the intraoperative frozen biopsy result was reported as positive, additional resection was performed and negative margin was achieved, or cTP was performed immediately. As a result, we selected the PD R0 group as the comparison target for the cTP R0 group and conducted an indirect comparison, because we aimed to evaluate whether the postoperative outcomes of cTP and PD were comparable to each other after R0 resection was achieved. PD R1 cases may be considered as a comparison target for the cTP R0 group. However, PD R1 cases were the cases where the SMA margin or retropancreatic margin was confirmed as tumor-positive in the final pathologic report after surgery. We thought those cases were not appropriate as a comparison target for the cTP R0 group in which radical clearance was obtained by confirming R0 after cTP.

Our study had limitations related to the small number of included subjects. This limitation is thought to be inevitable because TP is performed much less often than partial pancreatectomy such as PD or DP. Moreover, cases in which the surgery is converted to cTP during partial pancreatectomy due to intraoperative margin problems are even rarer. Although there was a small number of cases, we aimed to study the value of cTP performed to achieve R0 resection in a situation of intraoperative positive pancreatic parenchymal margin, because it is an important and difficult case. To overcome the limitation of small sample size, it would be effective to conduct a multi-center study involving several institutions in the future. We searched under retrospective design, and the data including morbidity and mortality were entirely based on the medical records of our center; therefore, our study has limitations related to this.

Despite these limitations, it is considered worthwhile to identify the proportion of cTP performed due to intraoperative positive pancreatic parenchymal margin at our high-volume center and study the postoperative short-term/long-term outcomes.

## 5. Conclusions

Even though radical clearance can be obtained after cTP in cases of intraoperative positive pancreatic parenchymal resection margin, postoperative long-term prognosis after cTP is not favorable. In such cases, careful consideration of the patients, such as general condition and other factors, is required when deciding to perform cTP.

**Author Contributions:** Conception/design: I.-W.H.; provision of study material or patients: J.-H.J.; collection and assembly of data: J.-H.J., S.-J.Y. and J.-S.H.; data analysis and interpretation: J.-H.J., O.-J.L., S.-H.S. and I.-W.H.; manuscript writing: J.-H.J. and I.-W.H. All authors have read and agreed to the published version of the manuscript.

**Funding:** This study was supported by Samsung Medical Center Grant #SMO1220431.

**Institutional Review Board Statement:** Our study was approved by the Institutional Review Board (IRB) of Samsung Medical Center (No. 2019-07-150). The IRB of Samsung Medical Center waived the need for consent from patients because our study was a retrospectively designed study. All methods were conducted in accordance with the relevant guidelines and regulations.

**Informed Consent Statement:** Informed consent was waived because it was approved to be harmless to all subjects.

**Data Availability Statement:** The datasets generated and analyzed during the current study are not publicly available due to the rules and regulations of our center but are available from the corresponding author on reasonable request.

**Acknowledgments:** All authors are sincerely grateful to all patients and surgeons who contributed to our study. All individuals included in our study consented to the acknowledgement.

**Conflicts of Interest:** The authors declare no conflict of interest.

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
