# Peer review of "Intraoperative Positive Pancreatic Parenchymal Resection Margin: Is It a True Indication of Completion Total Pancreatectomy after Partial Pancreatectomy for Pancreatic Ductal Adenocarcinoma?"

_curroncol, doi:10.3390/curroncol29080420_

Round 1

Reviewer 1 Report

Dr. Jung et al. evaluated the outcomes of total pancreatectomy (TP) as a result of additional pancreatic parenchymal resection for positive margin in the patients who initially intended partial pancreatectomy and showed poor survival even after R0 resection could be achieved.

The paper included clinically useful information suggesting the important careful patient selection indicated for TP as the authors concluded.

I have several concerns as the below.

1.       The most questionable issue in the clinical settings is whether TP should be performed to achieve R0 resection when we encounter positive pancreatic cut end margin during PD or DP. Were there any patients in PD group with R1 resection due to positive pancreatic cut end margin? Comparison of the outcomes between these patients and cTP group patients are expected.

2.       Was PD rather than DP initially indicated in all 25 patients in cTP group?

3.       Please present the number of patients with additional pancreatic resection as a result of positive margin in PD group.

4.       Generally, propensity score matching is performed using only the variable that could be evaluated only before intervention. Therefore, including pathologic TNM stage (but not clinical TNM), histopathologic factors, and postoperative adjuvant chemotherapy seems inappropriate. It would not matter if these histopathologic factors were different between the matched groups.

Reviewer 2 Report

I read with great interest the paper by Jung et al. on intraoperative margin assessment and the question if patients with an R1 resection at the transections margin after pancreatoduodenectomy (PD) should undergo total pancreatectomy (TP). 

The study includes patients from 1995 – 2015 with PD (R0+R1) or subsequent TP in cases of PD-R1 resections.  In total 1096 were assessed for illegibility. Of those, 632 underwent PD with R0 (445), R1 (177) and R2 resections (10). A total of 25 patients received TP after PD (R1) of which 21 had a negative margin (TP-R0) and 4 had a positive margin (TP-R1). 

The group conducted a propensity matching to compare the 21 individuals (TP-R0) with the control group of patients with PD-R0.  There were no differences regarding perioperatives outcomes and pathological findings between PD-R0 and TP-R0. However, the survival in patients with TP was significantly worse compared to those with PD-R0.

The study is of interest since the question if a resection (TP) in case of positive transection should be performed is still under debate. 

However, I have some questions for the authors:

-       It would be very interesting to compare the TP-R0 group (which have a worse survival compared to PD-R0) with cases of PD-R1. In addition, the authors should compare the survival between PD-R0 and PD-R1.

-       It would be very helpful if the authors could report the median survival in addition to the survival rates. 

-       In my opinion the authors should remove Table 5 as well as the survival analysis between TP-R0 and TP-R1 since the numbers are too small (4 patients with TP-R1) to draw any safe conclusions. 

Round 2

Reviewer 1 Report

The authors adequately responded to reviewer's comments. The remaining limitation were also well described in discussion section. I have no additional comments for further revision. 

Author Response

Comment and Suggestion for Authors :

The authors adequately responded to reviewer's comments. The remaining limitation were also well described in discussion section. I have no additional comments for further revision. 

Response to the reviewer :

We are very grateful and happy to receive your kind comments and encouragement. Thanks to you, we have a motivation to devote ourselves to research in the future.
